## PERSPECTIVES

# Gastric inhibitory polypeptide as the new candidate for the interaction of skeletal muscle blood flow and glucose disposal

Andrea Tamariz-Ellemann[1] (ID),
Hannah G. Caldwell[1,2]
and Lasse Gliemann[1] (ID)

[1]*The August Krogh Section for Human Physiology, Department of Nutrition, Exercise and Sports, University of Copenhagen, Copenhagen, Denmark*
[2]*Centre for Heart, Lung and Vascular Health, School of Health and Exercise Sciences, University of British Columbia Okanagan, Kelowna, BC, Canada*

Email: gliemann@nexs.ku.dk

Edited by: Kim Barrett & Bettina Mittendorfer

Linked articles: This Perspectives article highlights an article by Roberts-Thomson *et al*. To read this paper, visit https://doi.org/10.1113/JP282428.

The peer review history is available in the Supporting Information section of this article (https://doi.org/10.1113/JP282843#support-information-section).

## Introduction

The stimulated increase in skeletal muscle microvascular blood flow (MBF) observed in the postprandial state is imperative in the promotion of glucose disposal and hormone delivery to the myocyte. This vascular response is lost in the pathological state of insulin resistance, showcasing the importance of insulin to promote MBF. Therefore, to mimic the postprandial state, intravenous infusions of insulin and glucose are frequently used in the experimental setting; for example, hyperinsulinaemic euglycaemic clamp, leading to an increase in MBF. Hyperglycaemic meals, however, impair MBF, suggesting that the postprandial state is governed by mechanisms different from those of an experimental glucose challenge. Traditionally, the hormones that have received the most attention are insulin and GLP-1, but maybe the importance of a previously underappreciated incretin, gastric inhibitory polypeptide (GIP) has been overlooked.

Recent findings by Roberts-Thomson *et al.* (2022), published in this issue of *The Journal of Physiology*, contribute to this suggestion by revealing opposite vascular responses and related hormone release in response to glucose depending on *route* of administration.

Roberts-Thomson *et al.* (2022) compared the skeletal muscle micro- and macrovascular responses to an oral glucose tolerance test (OGTT) and an intravenous glucose tolerance test (IVGTT) in healthy individuals ($n = 10$ 4/6, M/F). The aim was to identify potential gut-derived hormones that could help explain the contradictory MBF response apparently dependent on the route of glucose administration. By use of contrast enhanced ultrasound to measure thigh muscle microvascular haemodynamics, MBF was correlated to measurements of plasma insulin and the incretins gastric inhibitory polypeptide (GIP) and glucagon-like peptide-1 (GLP-1) during oral and intravenous glucose administration. Consistent with their hypothesis, the results showed that the OGTT impaired MBF while IVGTT increased MBF. It is noteworthy that the plasma insulin concentration was approximately 300% higher in response to OGTT compared with IVGTT and the changes in GIP were negatively correlated with $\Delta$MBF while the GLP-1:GIP log ratio was positively correlated with $\Delta$MBF. Taken together, these findings indicate an incretin-stimulated regulation of MBF (via GIP) and emphasize the importance of *route* of glucose administration on MBF; these results are especially relevant for experimental study design. When administrating glucose orally, the *route* for glucose disposal involves the gut, in a way that corresponds with the glucose disposal observed in the true postprandial state. This pathway includes 'the incretin effect' on insulin promotion; however, new research indicates that GLP-1 and GIP stimulate glucose extraction independently of insulin in part via their vasoactive effects (see Fig. 1).

## Why and how does oral glucose impair skeletal muscle blood flow?

Overall, OGTT induces a release of GLP-1 and GIP that will also promote the release of insulin. Intra-arterial infusion of GLP-1 has been shown to increase muscle blood flow in humans (Sjøberg *et al.* 2014); this response is perhaps explained by GLP-1 activation of G-protein coupled receptors on endothelial cells, eventually leading to activation of endothelial nitric oxide synthase (eNOS) and release of the potent vasodilator nitric oxide (Dong *et al.* 2013). Collectively, the OGTT-induced increase in GLP-1 does not explain the reduction in skeletal muscle blood flow, and Roberts-Thomson and colleagues further show that there is no association between GLP-1 levels and MBF.

Interestingly, GIP was negatively correlated with MBF, suggesting that increased GIP levels are involved in the OGTT-induced vasoconstriction. GIP acts via its G-protein coupled receptors on the endothelial cells; however, this leads to elevated $Ca^{2+}$ and cAMP. Although higher cAMP levels increase eNOS activity, there is convincing evidence that increased endothelial $Ca^{2+}$ levels cause endothelin-1 (ET-1) release, thus – as the authors suggest – probably explaining the OGTT-induced vasoconstriction. While ET-1 levels were not measured in the current study, previous work substantiates that OGTT induces ET-1 release and that ET-1 receptor blockade blunts the GIP-induced vasoconstriction (Ding *et al.* 2003).

Insulin may also induce endothelial ET-1 release via the MAP kinase and thus could also be a player in the vasoconstriction observed with OGTT. However, insulin can also, via the PI3-K pathway, activate eNOS to release the vasodilator nitic oxide and thus oppose the constrictive effect of the ET-1 pathway. Hyperinsulinaemic euglycaemic clamps are characterized by *increased* MBF, with elevated plasma insulin levels up to $\sim$600 pmol/l (McConell *et al.* 2020) compared to the $\sim$300 pmol/l observed in the present study by Roberts-Thomson, and thus insulin can be ruled out as a candidate for mediating the vasoconstrictive effects of an OGTT.

The understanding of the involvement of GIP in the regulation of microvascular tone in skeletal muscle, is still in its infancy, but it appears that GIP is the hot new incretin that affects blood flow regulation and ultimately

*The Journal of Physiology*

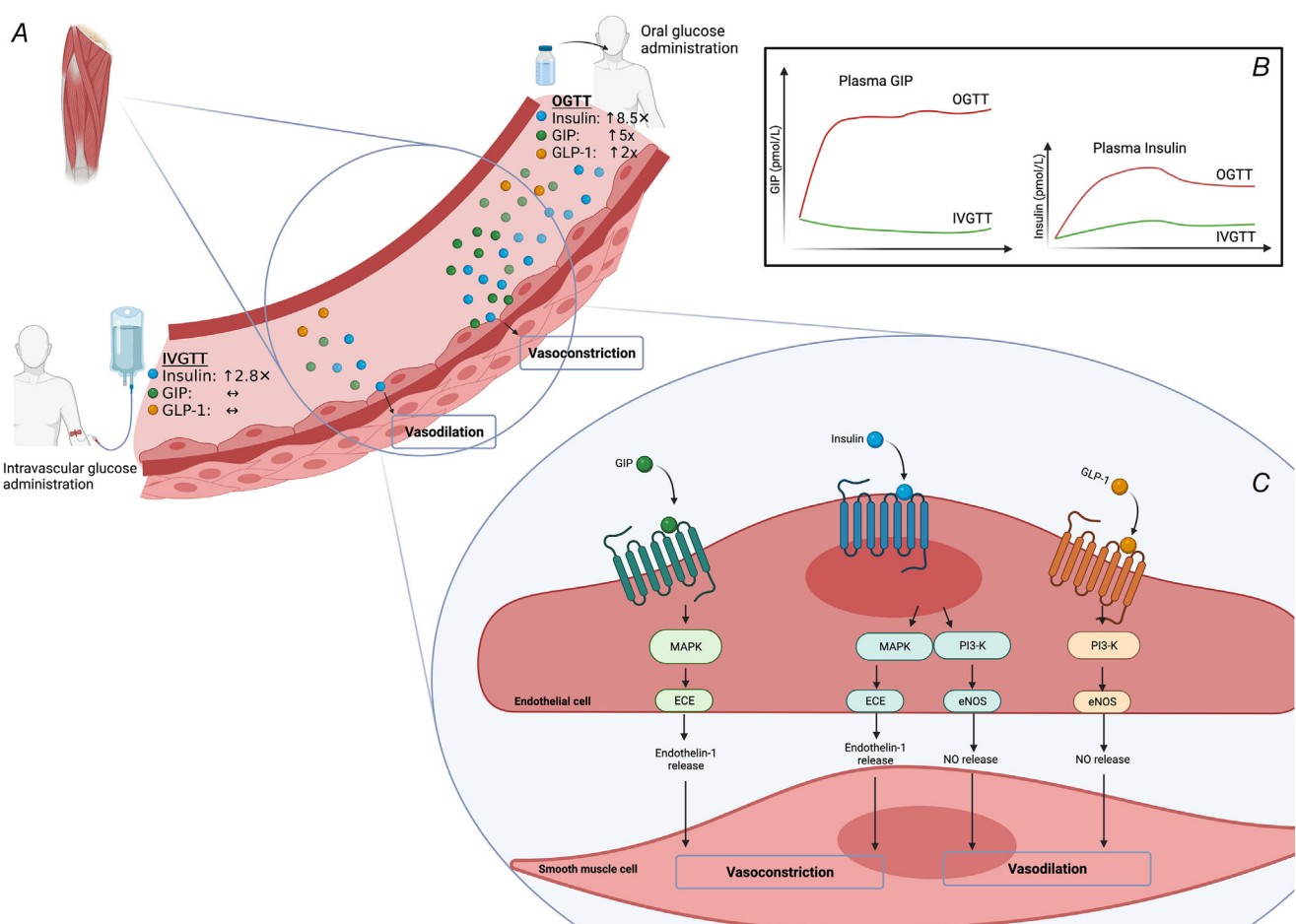

**Figure 1. Effect of oral vs. intravenous glucose on skeletal muscle blood vessels**
*A*, in skeletal muscle microvascular blood vessels, oral glucose tolerance test (OGTT) leads to vasoconstriction whereas intravenous glucose tolerance test (IVGTT) results in vasodilatation. *B*, this difference in vascular response to the same glucose load is probably related to the marked increased in gastric inhibitory polypeptide (GIP) and not dependent on the increase in plasma insulin or glucagon like peptide 1 (GLP-1). *C*, the proposed vasoconstrictive effect of GIP is the result of GIP-receptor dependent MAPK induced increase in endothelin-1 release via increases in endothelin converting enzyme (ECE) (Ding *et al.* 2003). GLP-1 acts via activation of G-protein coupled receptors, activating several intracellular signalling pathways via phosphoinositide 3-kinase (PI3K) to increase endothelial nitric oxide synthase (eNOS) phosphorylation and ultimately release of the vasodilator compound nitric oxide (NO). Insulin also acts via endothelial cell located G-protein coupled receptors and may induce both vasoconstriction via the endothelin-1 pathway or vasodilatation via increased eNOS activity.

glucose disposal. Moreover, GIP may be playing a pivotal role in the development of lifestyle-induced insulin resistance and impaired vascular function.

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

## Additional information

### Competing interests

None.

### Author contributions

All authors have approved the final version of the manuscript and agree to be accountable for all aspects of the work. All persons designated as authors qualify for authorship, and all those who qualify for authorship are listed.

### Funding

None.

### Keywords

gastric inhibitory polypeptide, hyperglycaemia, skeletal muscle

## Supporting information

Additional supporting information can be found online in the Supporting Information section at the end of the HTML view of the article. Supporting information files available:

**Peer Review History**

