## [Peer Review History · The Journal of Physiology]

Gastric Inhibitory Polypeptide as the new candidate for the interaction of skeletal muscle blood flow and glucose disposal

Andrea Tamariz-Ellemann, Hannah Caldwell, and Lasse Gliemann

DOI: 10.1113/JP282843

Corresponding author(s): Lasse Gliemann (gliemann@nxs.ku.dk)

Review Timeline:

Submission Date:	01-Feb-2022
Editorial Decision:	07-Feb-2022
Revision Received:	09-Feb-2022
Accepted:	10-Feb-2022

Senior Editor: Kim Barrett

Reviewing Editor: Bettina Mittendorfer

Transaction Report:

Dear Dr Gliemann,

Re: JP-P-2022-282843 "Gastric Inhibitory Polypeptide as the new candidate for the interaction of skeletal muscle blood flow and glucose disposal" by Andrea Tamariz-Elleemann, Hannah Caldwell, and Lasse Gliemann

Thank you for submitting your invited Perspectives article to The Journal of Physiology. It has been assessed by a Reviewing Editor and the author of the focus paper.

Minor alterations have been requested.

The reports are copied at the end of this email. Please address all of the points and incorporate all requested revisions.

NEW POLICY: In order to improve the transparency of its peer review process The Journal of Physiology publishes online as supporting information the peer review history of all articles accepted for publication. Readers will have access to decision letters, including all Editors' comments and referee reports, for each version of the manuscript and any author responses to peer review comments. Referees can decide whether or not they wish to be named on the peer review history document.

I hope you will find the comments helpful and have no difficulty in revising your article within 7 days.

To submit the revised version use the links in Author Tasks Link Not Available.

Please ensure that the article is a Word File with no more than 5 references, including the focus paper.

Thank you for your contribution to the Journal.

Yours sincerely,

Professor Kim E. Barrett
Editor-in-Chief
The Journal of Physiology
<https://jp.msubmit.net>
<http://jp.physoc.org>
The Physiological Society
Hodgkin Huxley House
30 Farringdon Lane
London, EC1R 3AW
UK
<http://www.physoc.org>
<http://journals.physoc.org>

EDITOR COMMENTS

Reviewing Editor:

The reviewers noticed some minor inaccuracies that need to be corrected before this perspective can be accepted.

REFEREE COMMENTS:

Referee #1:

This is a nice summary of the manuscript and includes a very detailed figure. Could the authors please make the following small changes:

1. Write the first authors name correctly throughout the text - it should say "Roberts-Thomson" and not "Robert-Thomson" or "Robert-Thomsen".
2. The paper was published in 2022 and not 2021. This should be changed throughout the text.
3. In the figure, the authors have indicated that GLP-1 did not change during the OGTT. GLP-1 was significantly elevated after the OGTT at 20 and 40 min. This change should be reflected in the figure.

REFEREE COMMENTS:

Referee #1:

This is a nice summary of the manuscript and includes a very detailed figure. Could the authors please make the following small changes:

1. Write the first authors name correctly throughout the text - it should say "Roberts-Thomson" and not "Robert-Thomson" or "Robert-Thomsen".
2. The paper was published in 2022 and not 2021. This should be changed throughout the text.
3. In the figure, the authors have indicated that GLP-1 did not change during the OGTT. GLP-1 was significantly elevated after the OGTT at 20 and 40 min. This change should be reflected in the figure.

We apologize for these errors and have made all corrections in the new version.

Dear Dr Gliemann,

Re: JP-P-2022-282843R1 "Gastric Inhibitory Polypeptide as the new candidate for the interaction of skeletal muscle blood flow and glucose disposal" by Andrea Tamariz-Ellemann
Hannah Caldwell
Lasse Gliemann

I am pleased to tell you that your invited Perspective article has been accepted for publication in The Journal of Physiology.

NEW POLICY: In order to improve the transparency of its peer review process The Journal of Physiology publishes online as supporting information the peer review history of all articles accepted for publication. Readers will have access to decision letters, including all Editors' comments and referee reports, for each version of the manuscript and any author responses to peer review comments. Referees can decide whether or not they wish to be named on the peer review history document.

The last Word version of the paper submitted will be used by the Production Editors to prepare your proof. When this is ready you will receive an email containing a link to Wiley's Online Proofing System. The proof should be checked and corrected as quickly as possible.

All queries at proof stage should be sent to tjp@wiley.com

Thank you very much for your contribution to The Journal of Physiology.

Yours sincerely,

Professor Kim E. Barrett
Editor-in-Chief
The Journal of Physiology
<https://jp.msubmit.net>
<http://jp.physoc.org>
The Physiological Society
Hodgkin Huxley House
30 Farringdon Lane
London, EC1R 3AW
UK
<http://www.physoc.org>
<http://journals.physoc.org>